# Fluorescence In Situ Hybridization in Primary Diagnosis of Biliary Strictures: A Single-Center Prospective Interventional Study

**DOI:** 10.3390/biomedicines11030755

**Published:** 2023-03-02

**Authors:** Vincent Dansou Zoundjiekpon, Premysl Falt, Jana Zapletalova, Petr Vanek, Daniela Kurfurstova, Zuzana Slobodova, Daniela Skanderova, Gabriela Korinkova, Pavel Skalicky, Martin Lovecek, Ondrej Urban

**Affiliations:** 12nd Department of Internal Medicine, Gastroenterology and Geriatrics, University Hospital Olomouc, Faculty of Medicine and Dentistry, Palacky University Olomouc, 779 00 Olomouc, Czech Republic; 2Department of Medical Biophysics, Faculty of Medicine and Dentistry, Palacky University Olomouc, 779 00 Olomouc, Czech Republic; 3Institute of Clinical and Molecular Pathology, University Hospital Olomouc, Faculty of Medicine and Dentistry, Palacky University Olomouc, 779 00 Olomouc, Czech Republic; 4Department of Surgery I, University Hospital Olomouc, Faculty of Medicine and Dentistry, Palacky University Olomouc, 779 00 Olomouc, Czech Republic

**Keywords:** primary diagnosis of biliary strictures, first retrograde cholangiopancreatography, brush cytology, fluorescence in situ hybridization, ZytoLight FISH probe

## Abstract

**Background and aims:** Diagnosis of the biliary stricture remains a challenge. In view of the low sensitivity of brush cytology (BC), fluorescence in situ hybridization (FISH) has been reported as a useful adjunctive test in patients with biliary strictures. We aimed to determine performance characteristics of BC and FISH individually and in combination (BC + FISH) in the primary diagnosis of biliary strictures. **Methods:** This single-center prospective study was conducted between April 2019 and January 2021. Consecutive patients with unsampled biliary strictures undergoing first endoscopic retrograde cholangiopancreatography in our institution were included. Tissue specimens from two standardized transpapillary brushings from the strictures were examined by routine cytology and FISH. Histopathological confirmation after surgery or 12-month follow-up was regarded as the reference standard for final diagnosis. **Results:** Of 109 enrolled patients, six were excluded and one lost from the final analysis. In the remaining 102 patients (60.8% males, mean age 67.4, range 25–92 years), the proportions of benign and malignant strictures were 28 (27.5%) and 74 (72.5%), respectively. The proportions of proximal and distal strictures were 26 (25.5%) and 76 (74.5%), respectively. In comparison to BC alone, FISH increased the sensitivity from 36.1% to 50.7% (*p* = 0.076) while maintaining similar specificity (*p* = 0.311). **Conclusions:** Dual-modality tissue evaluation using BC + FISH showed an improving trend in sensitivity for the primary diagnosis of biliary strictures when compared with BC alone.

## 1. Introduction

Many benign and malignant disorders, mostly of hepato-pancreato-biliary origin, manifest as biliary strictures. Differential diagnosis is often difficult. According to current literature, 70–80% of biliary strictures are malignant while 20–30% are benign [1,2]. Timely diagnosis is essential to enable relevant patient management, which includes surgery and oncological treatment in cases of malignancy and endoscopic treatment when benign etiology is proven [2,3].

When clinical and laboratory examination and cross-sectional imaging workup are unable to establish a definitive diagnosis, a tissue evaluation is required. Despite improvements in this field, tissue diagnosis remains challenging [4,5,6]. For tissue acquisition, endoscopic retrograde cholangiopancreatography (ERCP) with transpapillary brushing for cytology analysis is recommended as the first-line approach [2,3,7,8,9]. Although brush cytology (BC) is available and safe, limited sensitivity to malignancy in the range of 19–56% remains an issue [3,7,8,9,10,11,12]. Among several other factors, the extrinsic nature and low cellularity of some tumors play a role.

When imaging studies and standard transpapillary tissue sampling are nondiagnostic for a suspected malignant biliary stricture, the term “indeterminate biliary stricture” (IBS) is used [2,5]. In these cases, advanced endoscopy techniques, such as cholangioscopy with forceps biopsy or endoscopic ultrasonography (EUS) with fine-needle biopsy, are recommended [2,3,8,13,14,15,16,17]. Nevertheless, these methods are expert-dependent, costly, and not universally available [13,17]. Moreover, the risk of complications should be considered, although their reported incidence is low [17].

In order to increase the diagnostic yield of BC, several improvements have been recently suggested [8,12,18,19]. Among these, fluorescence in situ hybridization (FISH) has shown promising results [8,20]. FISH is a molecular cytogenetic method based on the detection of fluorescently labeled specific DNA/RNA sequences of the chromosomes with a high degree of sequence complementarity [21]. Typically, FISH using UroVysion^®^ probes enables the detection of aneuploidy for chromosomes 3, 7, and 17 and loss of the 9p21 in patients with suspected pancreatobiliary malignancy [20,22,23,24,25]. Several studies have demonstrated that FISH, when combined with routine BC, increased the overall sensitivity from the 21–50% range to 58–69% while maintaining high specificity in the IBS [8,20,24,25,26]. These studies nevertheless have some limitations, such as a retrospective design, limited number of patients, and short follow-up. 

We present the results of a single-center prospective interventional study aimed at determining the performance characteristics of BC and FISH individually and in combination in a primary diagnosis of biliary strictures.

## 2. Methods

This prospective trial was conducted within a single tertiary endoscopy center in Olomouc, the Czech Republic, from April 2019 to January 2021. It was approved by the local institutional review board (EK 15219) and registered at Clinicaltrials.gov (NCT04391153). Patients undergoing first ERCP for previously unsampled biliary stricture were included. Exclusion criteria were age <18 years, non-ability to provide signed informed consent, history of transpapillary or EUS stricture sampling, isolated intrahepatic biliary strictures, inaccessible major papilla, acute cholangitis, and significant coagulopathy. Tissue specimens obtained via transpapillary brushing during ERCP were examined by routine cytology methods. In addition, one brushing for FISH analysis was performed. After the procedure, patients were hospitalized for 24–72 h. Bleeding without the need for transfusion, Tokyo grade I cholangitis, and post-ERCP pancreatitis (PEP) with a short hospitalization (<4 days) were considered as mild complications. Bleeding needing blood transfusion, Tokyo grade II and III cholangitis, PEP with organ failure, or requiring longer hospitalization (≥4 days), as well as any perforation were considered as severe complications.

### 2.1. Tissue Acquisition

ERCP was performed by two experienced endoscopists using standard duodenoscopes (TJF-Q 180V or TJF-Q190V, Olympus, Hamburg, Germany) with patients under conscious sedation or general anesthesia. In all naive cases, biliary sphincterotomy was performed. Measures for the prevention of post-ERCP pancreatitis (PEP) and cholangitis were employed in accordance with available guidelines. 

During ERCP, each biliary stricture was sampled twice using the cytology brush (BrushMaster V, Olympus, Hamburg, Germany) inserted over a guidewire, with at least 10 to-and-fro motions through the stricture per sampling under fluoroscopy control. Dilatation of the stricture was not routinely performed. Material from the first sampling was smeared directly onto microscopic slides for routine cytology. After the second sampling, the brush was cut and placed into 10 mL of ThinPrep CytoLyt solution (Hologic, Marlborough, MA, USA) for subsequent FISH analysis. The sampling was considered successful when the tissue sample obtained was of sufficient quantity and quality to enable both cytological and FISH analysis. Routinely, biliary plastic or fully covered self-expandable metallic stents were introduced. 

### 2.2. Cytological and Molecular-Cytogenetic Analysis

#### 2.2.1. Cytological Analysis

After macroscopic evaluation of the samples, smeared microscopic slides were fixed in 96% ethanol. For the cytological evaluation, the smear was stained with May–Grünwald/Giemsa–Romanovsky and hematoxylin–eosin. Specimens were then analyzed by pathologists as a part of the standard clinical procedures. The six-tiered standardized terminology and nomenclature for pancreatobiliary cytology proposed by the Papanicolaou Society (*PAP categories I: non-diagnostic, II: negative for malignancy, III: atypical, IV: neoplastic benign or other, V: suspicious for malignancy, VI: positive for malignant*) were used [27]. For purposes of the study, PAP categories V and VI were considered as positive for malignancy while the categories I, II, III, or IV were regarded as nonmalignant. (Figure 1A,B).

#### 2.2.2. Fluorescence In Situ Hybridization

A brush placed in the 10 mL of ThinPrep CytoLyt solution was provided to the cytogenetics laboratory within 1 h of collection. Samples processing was started within 24 h. The fluid was centrifuged and harvested cells were fixed by freshly made 3:1 methanol-acetic acid. The cell pellets were placed onto the slides and left to dry. Slides were stored in a refrigerator. Further processing of the samples was a two-day process. In cases where the material was non-representative for FISH evaluation, the examination was repeated from the remaining cytology slides. FISH was performed according to the manual for the procedure using the commercially available ZytoLight FISH-cytology implementation kit and ZytoLight SPEC CDKN2A/CEN 3/7/17 quadruple color probe (ZytoVision^®^, Bremerhaven, Germany). This enables the detection of aneuploidy for chromosomes 3, 7, and 17 and loss of the 9p21, seen as different colors, as follows: chromosome 3 centromere enumeration probe (CEP 3) = red, CEP 7 = green, CEP 17 = blue, and 9p21/p16 = gold (Figure 2).

FISH evaluation was performed by two experienced pathologists who were blinded to BC results. The positivity of FISH for chromosomes 3, 7, and 17 was defined by the presence of polysomy of these chromosomes and the positivity of FISH for chromosomal region 9p21(p16) was defined by a presence of heterozygous deletion or homozygous deletion for 9p21. A homozygous deletion included absence of both copies of CDKN/2 (p16) genes in at least 10 cells. A heterozygous deletion included the absence of one of the genes in at least 6% of the total number of cells. Polysomy was defined by a gain of two or more chromosomes in at least five cells. FISH was negative in cases when polysomy or p16 deletion was found to be absent. The positivity was defined by the presence of polysomy and/or the presence of 9p21 deletion of two or more chromosomes. The FISH was inconclusive when there was polysomy or the presence of 9p21 deletion in only one chromosome. Specimens with an insufficient number of cells for FISH testing were considered as non-diagnostic by FISH and regarded as negative for FISH in the ensuing analysis. 

### 2.3. Follow-Up and Final Diagnosis

A histopathological report from the surgical resection was regarded as the gold standard for final diagnosis. Any decision to perform surgery was made within a multidisciplinary tumor board. Patients not indicated for surgery were followed for 12 months to confirm the stricture to be benign or malignant. The follow-up protocol consisted of a three-month periodic examination in a relevant laboratory, cross-sectional imaging methods, and repeated endoscopic procedures. In cases of pancreatic mass lesions causing extrinsic biliary strictures, EUS with fine-needle biopsy was performed. Patients were managed adequately based on the results and a multidisciplinary consensus during the follow-up period. An autopsy or reexamination of all clinical data and all available results was considered in the case of death.

### 2.4. Statistical Analysis

All statistical analyses were carried out using IBM SPSS Statistics version 23 (IBM Corp., Armonk, NY, USA). Sensitivity and specificity with 95% confidence interval (CI), as well as positive predictive value (PPV) and negative predictive value (NPV) were computed using definitive diagnosis (malignant/benign) as a reference method. A chi-square test was used to compare the sensitivity and specificity of the brush method and the combination of the brush and FISH methods. For all tests, a *p*-value of <0.05 was considered statistically significant.

## 3. Results

In total, 109 consecutive patients were enrolled in the study during the study period. Six patients with an already known diagnosis of primary sclerosing cholangitis (PSC) without suspicion of malignancy were excluded and one patient was lost from the follow-up (Figure 3). As a result, a total of 102 patients, including 62 (60.8%) males and mean age 67.4 (range 25–92) years, were subjected to final analysis. The demographic data of the patients and their definitive diagnosis are presented in Table 1 and Table 2. In 22 (21% of) cases, the strictures had to be dilated to enable the brushing. The sampling was judged macroscopically as successful in all cases. There were no lethal or severe complications related to the study methods. A total of three (2.8%) patients developed mild PEP.

Benign strictures were diagnosed in 28 (27.5%) patients and malignancy in 74 (72.5%) patients, including 25.5% with cholangiocarcinoma and 37.3% with pancreatic tumors. A total of three (2.9%) patients had primary sclerosing cholangitis (PSC), two of them with malignancy. A total of 24 (23.5%) patients were treated surgically, of whom 20 (83.3%) were diagnosed with malignancy (Table 1).

Operating characteristics of study methods are shown in Table 3, Table 4 and Table 5. Notably, FISH increased the sensitivity of BC from 36.1% to 50.7% (*p* = 0.076) in the entire cohort and from 48% to 69.2% in the cohort of patients with intrinsic strictures (*p* = 0.124). On the other hand, in the subgroup with extrinsic strictures the sensitivity of BC was only increased from 31.3% to 41.7% (*p* = 0.290). Taking into account the etiology of the stenosis, BC and BC + FISH had a sensitivity of 31.6% and 42.1% (*p* = 0.343), respectively, in biliary strictures caused by pancreatic mass lesions, compared to 45.8% and 68% (*p* = 0.117) respectively in those due to cholangiocarcinoma (Appendix A). The differences in specificity of the study methods in all subgroups were negligible.

Considering the significance of individual chromosome alterations diagnosed by FISH, the sensitivity of BC increased from 36.1% to 57.5% and 65.8% with a specificity of 74.1% and 57.1% for chromosomes 3 and 7, respectively, in the overall study population, compared to 80.8% and 76.9% for the sensitivity and 73.3% and 66.7% for the specificity in patients with intrinsic strictures (Appendix A). 

The FISH success rate was 71.6% and FISH was inconclusive in 29 (28.4%) cases in our study. Only in one case (1%) was the FISH material non-representative.

Appendix A shows the demographic parameters of the patients who died during the follow-up. During the study period, 11 (10.8%) patients died. This includes 10 (90.9%) with malignant strictures and one patient who was a 73-year-old male with chronic pancreatitis and a stenosis of the distal bile duct who died after six months of lung cancer (Table 1).

## 4. Discussion

The diagnosis of biliary stricture is challenging. Despite the progress in imaging methods, tissue diagnosis is required in most cases. Current guidelines recommend ERCP with BC as the first-line method [2,3,7,8,9]. While brushing is easy to perform and safe, the operating characteristics of BC are suboptimal, with sensitivity for malignancy only in the range of 19–56% [3,7,8,9,10,11]. As a result, after initial ERCP, many strictures remain indeterminate and utilization of advanced sampling techniques, either cholangioscopy or EUS-based, is recommended [2,3,8,14,15,16,17]. Nevertheless, their use is associated with adverse events, and their costs and availability also remain issues. Therefore, new methods are needed to increase diagnostic accuracy of BC [8,12,18,19]. 

In our study, slides for FISH evaluation were cell-sufficient in 101/102 (99%) of cases. We investigated whether the combination of routine cytology and FISH using ZytoLight FISH probes improved the diagnostic yield of brushing. When compared with BC alone, this dual modality for tissue examination showed a trend toward increasing sensitivity from 36.1% to 50.7% (*p* = 0.076). Not surprisingly, in cases of intrinsic strictures, both figures were higher, at 48% and 69.2% (*p* = 0.124), respectively. These findings are consistent with the results of some other studies. For instance, in their prospective study, Gonda et al. demonstrated an increased sensitivity from 32% to 51% and Chaiteerakij et al. from 38% to 50% [20,25]. In other retrospective series, Liew et al. showed an increase from 54% to 69% and Hans et al. from 43% to 82% [28,29]. We consider these results as clinically relevant for the potential to lower the patient diagnostic burden while requiring minimal effort from an endoscopy point of view.

The specificity of BC in our series was 85.2%. In most relevant studies, this figure was close to 100%, although Liew et al., for instance, reported 82.4% and Han et al. 96% [28,29]. We identified two main reasons for this difference. First, we regarded findings as malignant if occurring not only in cases of category VI but also of category V, in which 83% of the cytology findings were in fact malignant (Appendix A). That is in accordance with the literature, where the risk of malignancy of brushing specimens designated as Category V is in the range of 80–96% [30,31]. Second, as many as 27.5% of our cases had a benign stricture. In some other studies, very small numbers of true negative samples make the specificity estimates less reliable. Specificity of FISH in our study was 82.1% when inconclusive results were taken as negative. This result has been shown in other studies to be in the range of 54–100% [8,22,24,28,29,32,33].

FISH is a molecular cytogenetic method based on the detection of fluorescently labeled specific DNA/RNA sequences of chromosomes with a high degree of sequence complementarity [21]. Typically, it enables the detection of various types of cytogenetic alterations (including aneusomy, aneuploidy, deletion, translocation, among others) for well-defined chromosomes in patients with suspected malignancies. In our study, FISH was performed using the ZytoLight FISH probes targeting chromosomes 3, 7, 17, and 9p21, which are the same chromosomes as those targeted by UroVysion probes used in other studies [20,25,28,29]. Even though these ZytoLight FISH probes were initially developed for the diagnosis of other neoplasms, such as hematologic, breast, lung, or urinary bladder cancers [34], our study showed that they could also be effective for the diagnostics of pancreatobiliary malignancies. Combined with routine cytology, they showed an improving sensitivity in the primary diagnosis of biliary strictures when compared to BC alone (*p* = 0.076). Although a set of probes (targeting the chromosomes 1q21, 7p12, 8q24, and 9p21) designed for pancreatobiliary malignancy and with 93% sensitivity and 100% specificity has recently been developed [24], it is not yet commercially available.

It would appear that next-generation sequencing (NGS) constitutes another alternative molecular cytogenetic method that could significantly increase the sensitivity of BC. Nevertheless, no diagnostic panel of mutations that would be typical for pancreatobiliary malignancy has yet been defined. The most common alterations are mutations of the genes p53, KRAS/NRAS, CDKN2A, SMAD 4, PTEN, and others [35,36,37]. Harbhajanka et al. showed in their publication that when combined with the BC, NGS significantly increased the sensitivity from 49% to 93% in premalignant lesions [36]. That, however, is currently a financially costly and time-consuming method for most laboratories, and this lost time could significantly influence the timely diagnosis and adequate management in patients with biliary strictures. Compared to NGS, FISH is a less costly, faster, and more widely available method.

Outside of our study protocol, according to the preference of the endoscopist, forceps biopsy was performed in 44 (43.1%) cases. The sensitivities of biopsy and combination of BC plus biopsy in this cohort were 40.6% and 66.7% (*p* = 0.002) while specificity was 91.7% and 73.3% (*p* = 0.347), respectively (Appendix A). As a result, forceps biopsy can be considered as an important adjunct to BC that is especially useful in cases of intrinsic strictures, as shown in the literature [5,8,17,29,38,39,40,41,42,43,44,45,46,47,48]. According to our out-of-protocol results, the sensitivity of BC + biopsy is higher than the sensitivity of BC + FISH (66.7% versus 50.7%). Additionally, biopsy and BC are financially less expensive than FISH. Nevertheless, biopsy may not always be technically successful, and its diagnostic yield could be affected by many factors [42,43,44,49], with many cases of false negativity for biopsy + BC. In these cases of questionable diagnosis, FISH could play an important role.

In some cases, the combination of BC, forceps biopsy and FISH could increase the sensitivity of biliary strictures as shown in Appendix A and in the literature [32]. However, this was not the aim of this study.

The strengths of our study include its prospective design, inclusion of naïve or previously unsampled biliary strictures, standardized methodology, definition of reference standards for final diagnosis, and follow-up completed in 99% of those patients included. On the other hand, a limited number of cases and the single-center design could be considered as limitations.

## 5. Conclusions

In comparison with BC alone, the combination of BC and FISH improved the sensitivity of biliary stricture diagnostics from 36.1% to 50.7%. FISH can be considered as a useful method for discrimination of biliary stricture etiology. Data from a larger prospective multicenter cohort of patients are needed.

## Figures and Tables

**Figure 1 biomedicines-11-00755-f001:**
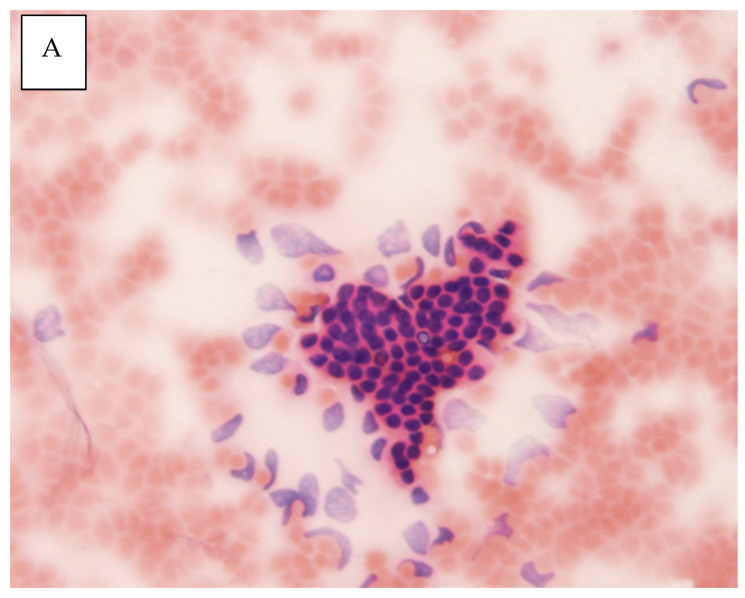
(**A**): A tuft of benign ductal epithelial cells with a regular arrangement of polarized uniform nuclei. Hematoxylin–eosin staining (100×). PAP II; (**B**): A cluster of malignant cells with irregular hyperchromic congested, often angular nuclei with irregular karyomembrane. Hematoxylin–eosin staining (100×). PAP VI.

**Figure 2 biomedicines-11-00755-f002:**
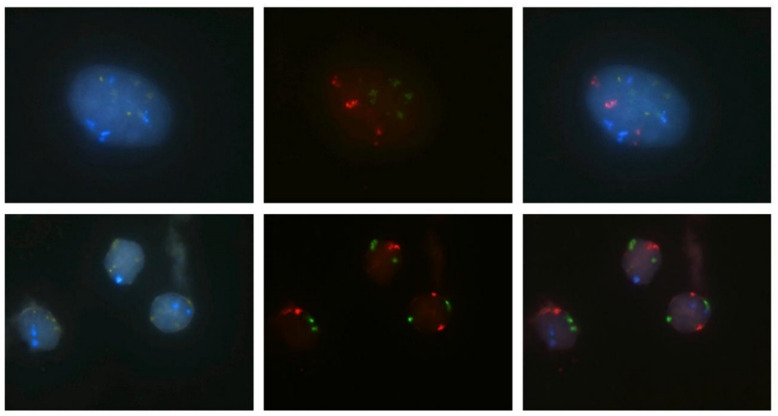
**Upper part of of this figure (positive FISH):** Nucleus of a tumor cell with increased number of red, green, and blue signals indicating polysomy of the chromosomes 3, 7, and 17, respectively. The third image summarizes the colors of all signals; round yellow (gold) signals marking CDKN2 (p16) are clearly visible, so deletion of this gene is not present; **Lower part of this figure (negative FISH):** Three nuclei of benign cells are visible without increased number of signals for chromosomes 3, 7, and 17, respectively, and without CDKN2(p16) deletion.

**Figure 3 biomedicines-11-00755-f003:**
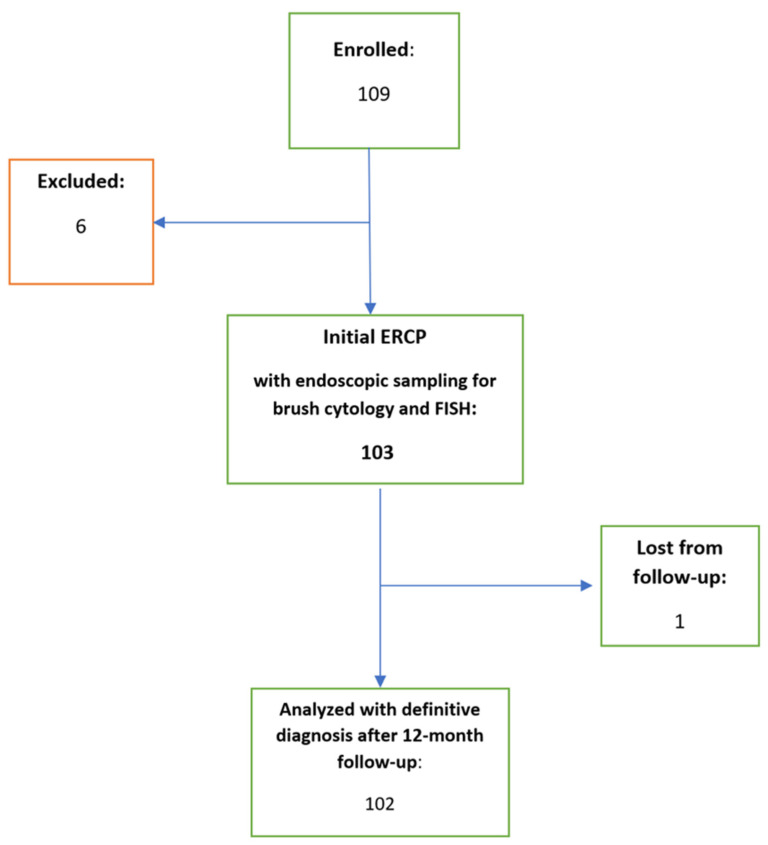
Study flow chart.

**Table 1 biomedicines-11-00755-t001:** Patient demographics.

Parameter	Study Cohort *N* = 102	Malignant Outcome *N* = 74	Nonmalignant Outcome *N* = 28
Age (mean) y	67.4 (25–92)	68.9 (50–92)	63.4 (25–89)
Gender (male), *n* (%)	62 (60.8%)	44 (59.5%)	18 (64.3%)
PSC *, *n* (%)	3 (2.9%)	2 (2.7%)	1 (3.6%)
Non-PSC, *n*	99 (97.1%)	72 (97.3%)	27 (96.4%)
Proximal stenosis, *n* (%)	26 (25.5%)	23 (31.1%)	3 (10.7%)
Distal stenosis, *n* (%)	76 (74.5%)	51 (68.9%)	25 (89.3%)
Intrinsic stricture, *n* (%)	40 (39.2%)	26 (35.1%)	14 (50%)
Extrinsic stricture, *n* (%)	62 (60.8%)	48 (64.9%)	14 (50%)
Surgical treatment, *n*	24	20	4
Endoscopic therapy, *n*	73	49	24
Oncological therapy, *n*	13	13	0
Combination therapy, *n*	8	8	-
Death, *n*	11	10	1

Endoscopic therapy = self-expanding metallic stents (SEMS), Radiofrequency ablation = RFA, One or multiple plastic stents, RFA + SEMS. Oncological therapy = adjuvant therapy, palliative therapy. Combination therapy = (two or more therapies—surgical or endoscopic or oncological).* PSC = primary sclerosing cholangitis.

**Table 2 biomedicines-11-00755-t002:** Study’s final clinical diagnosis after conclusive 12-month follow-up and/or postoperative histology.

**Biliary strictures** ***n* = 102**		**Final Clinical Diagnosis**	***n* (%)**
	Malignant *n* = 74	Proximal cholangiocarcinoma	17 (16.7%)
		Distal cholangiocarcinoma	9 (8.8%)
		Pancreatic tumors	38 (37.3%)
		B cells malignant/Lymphoma	3 (2.9%)
		Others (M)	7 (6.9%)
	Benign *n* = 28	Chronic pancreatitis	8 (7.8%)
		Choledocholithiasis	13 (12.7%)
		PSC	1 (1.0%)
		Others (B)	6 (5.9%)

Pancreatic tumors = pancreatic ductal adenocarcinoma, pancreatic lymphoma and malignant pancreatic cystic tumors; Others (M) = gallbladder adenocarcinoma, ampullary adenocarcinoma, metastatic disease (liver or malignant lymph nodes compression); Choledocholithiasis = stenosis caused by chronic presence of bile duct stones; PSC = primary sclerosing cholangitis; Others (B) = Iatrogenic biliary injuries, stenosis by reactive benign lymph nodes.

**Table 3 biomedicines-11-00755-t003:** Statistical performances of FISH, brush cytology (BC), and BC + FISH (overall study cohort).

*n* = 102	FISH	BC	BC + FISH	*p** (BC vs. BC + FISH)
Sensitivity (95% CI)	0.311 (0.208–0.429)	0.361 (0.251–0.483)	0.507 (0.387–0.626)	0.076
Specificity (95% CI)	0.821 (0.631–0.939)	0.852 (0.663–0.958)	0.741 (0.537–0.889)	0.311
PPV	0.821	0.867	0.841	1.000
NPV	0.311	0.333	0.357	0.781
Accuracy (95% CI)	0.451 (0.352–0.553)	0.495 (0.393–0.597)	0.570 (0.467–0.669)	0.289

FISH = fluorescence in situ hybridization, BC = brush cytology, PPV = positive predictive value, NPV = negative predictive value, CI = confidence interval, *p** = *p*-value (BC vs. BC + FISH).

**Table 4 biomedicines-11-00755-t004:** Statistical performances of brush cytology (BC) vs. BC + FISH (intrinsic stenosis group).

*n* = 40	BC	BC + FISH	*p*
Sensitivity (95% CI)	0.480 (0.278–0.687)	0.692 (0.482–0.587)	0.124
Specificity (95% CI)	0.867 (0.595–0.983)	0.733 (0.449–0.922)	0.651
PPV	0.857	0.818	1.000
NPV	0.500	0.579	0.600
Accuracy (95% CI)	0.625 (0.458–0.773)	0.707 0.454–0.839	0.432

PPV = positive predictive value, NPV = negative predictive value, CI = confidence interval, *p* = *p*-value, BC = brush cytology, FISH = fluorescence in situ hybridization.

**Table 5 biomedicines-11-00755-t005:** Statistical performances of brush cytology (BC) vs. BC + FISH (extrinsic stenosis group).

*n* = 62	BC	BC + FISH	*p*
Sensitivity (95% CI)	0.313 (0.187–0.462)	0.417 (0.276–0.568)	0.290
Specificity (95% CI)	0.833 (0.516–0.979)	0.750 (0.428–0.945)	0.617
PPV	0.882	0.870	0.910
NPV	0.233	0.243	0.917
Accuracy (95% CI)	0.417 (0.291–0.551)	0.483 (0.352–0.616)	0.467

PPV = positive predictive value, NPV = negative predictive value, CI = confidence interval, *p* = *p*-value, BC = brush cytology, FISH = fluorescence in situ hybridization.

## Data Availability

The data presented in this study are available on request from the corresponding author. The data are not publicly available due to privacy restrictions.

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
