# Peer review of "Fluorescence In Situ Hybridization in Primary Diagnosis of Biliary Strictures: A Single-Center Prospective Interventional Study"

_biomedicines, 2023, doi:10.3390/biomedicines11030755_

Round 1

Reviewer 1 Report

The article evaluated the utility of FISH for brush cytology (BC) specimens for biliary stricture. Unfortunately, FISH or FISH+BC did not have higher sensitivity than BC alone. In addition, BC+biopsy had the highest sensitivity and seems better method than FISH evaluation. However, FISH for BC specimens is a novel and interesting method. I have some questions as follows.

#1 Usually, FISH is costly and time-consuming. How long does it take to get results? 

#2 FISH requires sufficient specimens, and sometimes FISH is difficult to perform. Could all cases be evaluated FISH? Please show the success rate of FISH in this study.
In addition, please show the characteristics of the successful FISH case such as a comparison of the number of collected cells in BC specimen.

#3 In this study, BC+Biopsy was more useful than BC+FISH. Biopsy is cheaper and faster than FISH, therefore the gold standard for diagnosis seems to be still BC+Biosy.
However, FISH may also be useful for non-diagnostic cases. Please show the result of BC+Biopsy+FISH.

#4 Does Figure 1A show benign (PAP Category I)? Does 1B indicate PAP Category V?
Figure legend is necessary for Figure 1.

 #5 Please add the adverse events rate of the endoscopic procedure.

 #6 I do not understand the meaning of Table 4. Aren't all cases evaluated for bile duct stenosis? It seems misleading and unnecessary.

 #7 Half of the malignant cases were pancreatic tumors. The sensitivity of BC for pancreatic tumors is usually lower than for biliary tract cancer (BTC). This background may reduce the sensitivity of BC and FISH. Please show the result of the sub-group analysis for each etiology.

Author Response

Please see the attachement,

Best regards

Vincent Zoundjiekpon

Reviewer 2 Report

In the current manuscript, the authors share their experience on using two methods for biliary stricture analysis. 

In my opinion, the article is well written, with some minor comments:

- a high number of patients with pancreatic cancer were included. These type of patients do not require brushing and certainly the main method for the first diagnosis was not brushing. It is mentioned that a follow-up period was instated however, pancreatic cancer has a very low survival rate. It is mentioned that 11 patients died, but no data on regarding which type of patients. 

I would recommend comparing the pancreatic cancer patients with others and mentioning if you recommend this technique for these types of patients. 

The follow-up paragraph should be reformulated because you inserted many patients with pancreatic cancer and there is no mention of how their final diagnosis was made. 

You mention limitations only a small number of patients and the fact the study takes place only in a single center. There are many flaws, especially in patient selection and many others. Please include them.

Author Response

Best regards

Vincent Zoundjiepon

Round 2

Reviewer 1 Report

The author has well responded to my questions; however, some responses are different from what I intended.

Response to #3

I understand BC+biopsy is not the purpose of this study.

However, the sensitivity of BC+FISH is not higher than BC+biopsy.

Because FISH is an expensive examination than a biopsy and sensitivity is lower than BC+biopsy, the author needs to recommend a suitable case for FISH evaluation (ex. BC+biopsy false negative case).

Response to #6

I understood the meaning of Table 4-5 (the intrinsic stenosis group showed high sensitivity than the extrinsic stenosis group).

Please add the P-value in Table 5.

Page7. On the other hand,...lower (31.3 % a 41.7%) -> (31.3 % to 41.7%)

Response to #7

My question is not to show the subgroup of stricture location.

Please show the subgroup analysis of the sensitivity of each etiologies (BTC, pancreatic cancer, etc...),

Author Response

please see the attachement

Reviewer 2 Report

The authors took into consideration my recommendations and performed the necessary changes.

Author Response

Thank you very much for your review/ remarks

Sincerelly

Zoundjiekpon et al.